# The "DDVF" motif used by viral and bacterial proteins to hijack RSK kinases mimics a short linear motif (SLiM) found in proteins related to the RAS-ERK MAP kinase pathway

Martin Veinstein[1], Vincent Stroobant[2], Fanny Wavreil[1], Thomas Michiels [ID][1]*, Frédéric Sorgeloos[1,3]*

**1** de Duve Institute, Université catholique de Louvain, Brussels, Belgium, **2** Ludwig Institute for Cancer Research, Brussels, Belgium, **3** Centre Armand-Frappier Santé Biotechnologie, Institut National de la Recherche Scientifique, Laval, Québec, Canada

* frederic.sorgeloos@inrs.ca (FS); thomas.michiels@uclouvain.be (TM)

## Abstract

Proteins of pathogens such as cardioviruses, Kaposi sarcoma-associated herpes virus, varicella zoster virus and bacteria of the genus *Yersinia* were previously shown to use a common "DDVF" (D/E-D/E-V-F) short linear motif (SLiM) to hijack cellular kinases of the RSK (p90 ribosomal S6 kinases) family. Notably, the leader (L) protein of Theiler's murine encephalomyelitis virus (TMEV), a cardiovirus, and protein YopM of *Yersinia* species were shown to act as adapters to retarget RSKs toward unconventional substrates, nucleoporins and pyrin, respectively. Remarkable conservation of the SLiM docking site targeted by pathogens' proteins in RSK sequences suggested a physiological role for this site. Using SLiM prediction tools and AlphaFold docking, we screened the human proteome for proteins that would interact with RSKs through a DDVF-like SLiM. Co-immunoprecipitation experiments show that two candidates previously known as RSK partners, FGFR1 and SPRED2, as well as two candidates identified as novel RSK partners, GAB3 and CNKSR2 do interact with RSKs through a similar interface as the one used by pathogens, as was recently documented for SPRED2. FGFR1 employs a DSVF motif to bind RSKs and phosphorylation of the serine in this motif slightly increased RSK binding. FGFR1, SPRED2, GAB3 and CNKSR2 act upstream of RSK in the RAS-ERK MAP kinase pathway. Analysis of ERK activation in cells expressing a mutated form of RSK lacking the DDVF-docking site suggests that RSK might interact with the DDVF-like SLiM of several partners to provide a negative feed-back to the ERK MAPK pathway. Moreover, after TMEV infection, ERK phosphorylation was altered by the L protein in a DDVF-dependent manner. Taken together, our data suggest that, in addition to retargeting RSKs toward unconventional substrates, pathogens' proteins carrying a DDVF-like motif can compete with endogenous DDVF-containing proteins for RSK binding, thereby altering the regulation of the RAS-ERK MAP kinase pathway.

**Data availability statement:** All data are contained in the article or in the public repositories referred to in the article. All structural predictions and associated accuracy metrics have been made available in the zenodo open research data repository (Parts 1 and 2: https://doi.org/10.5281/zenodo.10630296, Parts 3 and 4: https://doi.org/10.5281/zenodo.10653846 and part 5: https://doi.org/10.5281/zenodo.10658284). The following protein accession numbers–Q7Z699, Q7Z698, Q2MJR0, Q13480, Q9UQC2, Q8WWW8, Q2WGN9, P11362, P21802, P22607, P22455, Q969H4, Q8WXI2, Q6P9H4, Q15418, P51812, Q15349, and Q9UK32-mentioned in the text are available in the UniProt database (https://www.uniprot.org). Biological material is available upon request.

**Funding:** MV was the recipient of a fellowship from the EOS joint programme of Fonds de la recherche scientifique-FNRS and Fonds wetenschappelijk onderzoek-Vlaanderen-FWO (EOS ID: 30981113 and 40007527), and then of an Aspirant fellowship from the Fonds de la recherche scientifique-FNRS. FS was supported by a contract of Scientific collaborator with the FNRS and then by a PDR grant (PDR T.0154.23) of the FNRS. Work was supported by FNRS (PDR T.0154.23), EOS (EOS ID: 30981113 and 40007527), and Loterie Nationale through support to the de Duve Institute. The funders had no role in study design, data collection and analysis, decision to publish, or preparation of the manuscript.

**Competing interests:** The authors have declared that no competing interests exist.

## Author summary

Short linear motif (SLiM) are 3 to 12 amino acid-long protein sequences that mediate the interaction with other proteins. We previously observed that highly unrelated pathogens, including viruses and bacteria, convergently evolved to hijack cellular enzymes of their host, through a common SLiM. In this work, we tested the hypothesis that the SLiM found in proteins of pathogens evolved to mimic a SLiM found in human proteins that regulate the cellular enzymes through the same interface. Protein-protein interactions mediated by SLiMs are often, low-affinity, transient interactions that are difficult to detect by conventional biochemical methods but that can nowadays be predicted with increasing confidence by artificial intelligence-based methods such as AlphaFold. Using such predictions, we identified several candidate human proteins and we confirmed experimentally that these proteins interact with the cellular enzymes the same way as pathogens' proteins do. Identified proteins belong to the well-known RAS-ERK MAPK pathway which regulates important functions of the cell, suggesting that pathogens evolved to hijack this MAPK pathway by SLiM mimicry. By doing so, they can both dys-regulate cellular physiology and hijack cellular enzymes to their own benefit.

## Introduction

As obligate cell parasites, viruses have evolved a number of ways to exploit or to interfere with key pathways in the cell, including the mitogen-activated protein (MAP) kinase pathways, in order to promote their own replication or to evade immune responses [1,2]. RSKs (p90 Ribosomal S6 Kinases) form a family of four closely related serine/threonine kinases which act downstream of the RAS-ERK (Rat sarcoma virus/ Extracellular signal-Regulated Kinase) MAP kinase pathway and regulate essential cellular processes such as growth, proliferation, survival, motility, and immunity [3]. Since these kinases play pivotal roles in cell physiology, tight regulation of their activity is essential [4–7] and this signaling pathway, including that of RSK kinases, has been shown to be dysregulated in developmental disorders, cancers, and pathogenic infections (RAS-ERK-MAPK: [8–10], RSKs: [11–15]). To ensure tight regulation, serine/threonine kinases commonly employ surface-located amino acids to interact with partners. These partners can be preferential substrates of the kinase or regulatory proteins that modulate kinase activity through catalysis, relocalization, or allostery [16]. The recruitment of binding partners may occur via short linear motifs (SLiMs) usually present in disordered regions of protein sequences. SLiMs play a significant role in the regulation of the human interactome [17], and the disordered regions from where they arise are mutated in more than 20% of human pathologies [18]. However, their low affinity and amino acid sequence degeneracy make them challenging to identify. To predict SLiM binding in docking grooves, Verburgt *et al.* and others have shown that AlphaFold-multimer [19] exhibits remarkable performance, allowing to predict potential interactors at docking sites [20,21].

SLiMs, which easily arise in viral protein sequences due to their rapid evolution, can be exploited by viruses as subversion strategies [22,23]. We previously reported that unrelated pathogens, such as RNA and DNA viruses (cardioviruses and herpesviruses) as well as bacteria (from the genus *Yersinia*), convergently evolved to use a short linear D/E-D/E-V-F motif (further referred to as "DDVF") to hijack host protein kinases of the RSK family [13,14,24]. The DDVF motif encoded by these pathogens interacts with a region encompassing the KAK-LGM residues, situated in a surface-exposed loop of the RSK kinases. Surprisingly, although

surface-exposed loops and pathogen-targeted protein regions often undergo rapid evolution, the KAKLGM residues of the SLiM-docking site remained remarkably conserved among all four isoforms of human RSK as well as across evolution. This suggests that microbial proteins evolved to target an essential RSK regulation site and raises the possibility that some cellular proteins might interact with RSK through the same interface [13,14]. This work thus aimed at identifying such cellular RSK partners to test whether pathogens thus used DDVF-mediated SLiM mimicry to dysregulate the RAS-ERK pathway.

## Results

### 1. Screening for candidate human proteins that bind RSKs through a DDVF motif

Previous findings showed that some proteins from pathogens evolved a DDVF motif that binds and regulates RSKs, effectively subverting a well-conserved region of the kinase. This prompted us to explore whether cellular proteins might interact with RSKs using the same interface [13]. To this end, we screened the human proteome for the presence of D/E-D/E-V-F motif using SlimSearch4 (http://slim.icr.ac.uk/slimsearch/) [25] and found 222 hits including several known RSK partners (Tab A in S1 Table). These protein candidates were further probed for RSK binding through AlphaFold-multimer machine learning structure prediction [19]. To this end, peptides encompassing the DDVF motif were in silico-folded with a fragment from the human RSK2 N-terminal kinase domain [21]. Out of 222 proteins tested, 67 were predicted, by at least one AlphaFold model to bind RSK2 in the KAKLGM docking groove in a configuration similar to the ORF45 protein encoded by KSHV [14] (Tab A in S1 Table). Of these, twelve proteins were described as known RSK interactors according to Biogrid and IntAct interaction databases, albeit with unknown binding interfaces [26,27]. Among DDVF-containing proteins, 9 out of 12 (75%) were known RSK partners and docked in RSK according to AlphaFold prediction. Among proteins with unknown ability to bind RSK, a lower proportion (67 out of 210 = ~31%) docked in RSK. This enrichment of successful predictions for known interactors supports the usefulness of AlphaFold screens in predicting protein-protein interactions.

### 2. SPRED2 and GAB3 DDVF motifs mediate RSK interaction

Among the 67 predicted RSK-interacting proteins identified above, we identified SPRED2 and GAB3 as candidates carrying the DDVF motif in a putative disordered region predicted to dock on RSK by AlphaFold-multimer in 5 out of 5 models (Tab A in S1 Table). Despite their location in an unstructured region typically prone to variation, the DDVF motifs of SPRED2 and GAB3 are remarkably conserved throughout evolution supporting the assumption that they are functionally important (Fig 1A and 1B). Despite SPRED2 showing a high degree of sequence similarity with its isoforms, the DDVF motif present in SPRED2 is not found in SPRED1 or SPRED3. Similarly, the DDVF motif in GAB3 is absent in GAB1, GAB2, and GAB4 (S1A and S1B Fig).

We used co-immunoprecipitation experiments to probe for an interaction of SPRED2 and GAB3 with RSK1. Controls included DDVF-to-DDVA mutations in the SLiM of SPRED2 and GAB3, as well a KAKLGM-to-KSEPPY mutation in the SLiM docking site of RSK1. Both types of mutations were previously shown to dramatically reduce the interaction between microbial proteins and RSKs [13]. As shown in Fig 1C–D, HA-RSK1 unambiguously co-immunoprecipitated with WT but hardly if at all with mutant FLAG-SPRED2 and FLAG-GAB3. Accordingly, HA-RSK1 co-immunoprecipitation was minimal with FLAG-SPRED1, which lacks a DDVF motif. Moreover, WT much more than mutant RSK1

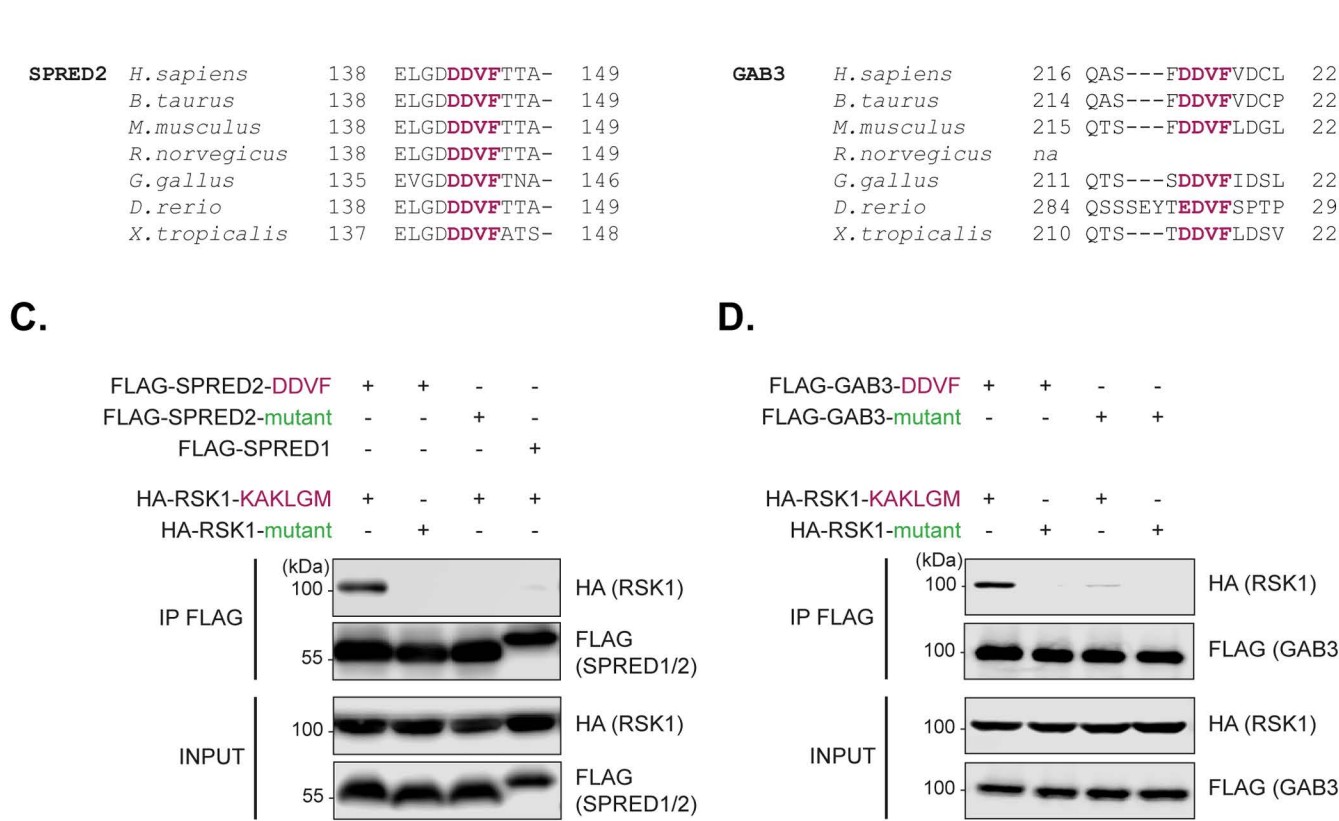

**Fig 1. Highly conserved DDVF motifs in SPRED2 and GAB3 mediate the interaction with RSK1. A-B.** Conservation of SPRED2 (A) and GAB3 (B) DDVF motifs (purple) across evolution **C-D.** Immunoblots showing the detection of wild type (KAKLGM) or mutated (KSEPPY) HA-RSK1 and FLAG (SPRED or GAB3 variants) in lysates (INPUT) of transfected HEK293T cells or after co-immunoprecipitation (IP FLAG) with various FLAG-SPRED variants (SPRED2-DDVF, SPRED2-DDVA mutant, or SPRED1), (n = 3) (C) or FLAG-GAB3 variants (GAB3-DDVF or GAB3-DDVA mutant), (n = 2) (D).

co-immunoprecipitated with FLAG-SPRED2 and FLAG-GAB3, suggesting that these proteins interact with RSK through the same DDVF:KAKLGM interface as the one used by microbial proteins. While this work was in progress, an elegant work by Lopez *et al.* [6] also showed the interaction of SPRED2 and RSK and provided the crystallographic structure of the DDVF:RSK interface. This structure largely confirmed the similarity of the interface formed between RSK and SPRED2 and that formed between RSK and pathogens' proteins, with the phenylalanine of the DDVF motif inserting in a hydrophobic pocket of RSK, at the level of the KAKLGM sequence [6].

## 3. Interaction of FGFR1 with RSK involves a DSVF motif which is subjected to regulation by phosphorylation

Our data [13], and those of Alexa *et al.* [14] and Lopez *et al.* [6] suggested a critical role of the VF residues within the DDVF motif for RSK binding but also of nearby acidic residues thought to increase affinity through electrostatic interactions. The presence of two aspartic residues surrounding the VF motif were not absolutely essential, one being sufficient for binding. We thus broadened our human proteome screening approach to accommodate these variations in the DDVF motif and screened the human proteome for the presence of either

the D/E-x-V-F or the D/E-V-F motif. This screening identified 1526 human proteins (Tab B in S1 Table) including 48 previously identified RSK interactors. Among these proteins, all four isoforms of Fibroblast Growth Factor Receptor (FGFR1-4) contained an evolutionarily well-conserved DSVF motif in their otherwise poorly conserved C-terminal tails (16 out of 61 conserved residues) (Figs 2A and S2A). Given that a previous study reported an interaction between RSK2 and the C-terminal tail of the FGFR1 receptor [4], we sought to investigate whether this DSVF motif could mediate the interaction of FGFR1 with the KAKLGM site of RSK.

The DSVF motif of all FGFR isoforms successfully docked into the RSK KAKLGM site in 5 out of 5 AlphaFold-multimer predictions (see S1C Fig and Tab B in S1 Table). To experimentally assess the importance of the DSVF motif for the RSK:FGFR1 interaction, we conducted co-immunoprecipitation assays in HeLa cells co-transfected with plasmids expressing HA-FGFR1 (DSVF or DSVA mutant) and FLAG-RSK1 (wild type KAKLGM or KSEPPY mutant) (Fig 2B). Our data show that the DSVF motif of FGFR1 indeed interacts with the KAKLGM site of RSK1. As Nadratowska-Wesolowska *et al.* [4] demonstrated that FGFR1-RSK2 interaction depends on MAPK activation, we pretreated cells with PMA before co-immunoprecipitation. However, this did not significantly increase the association of RSK1 with FGFR1 in our experiments.

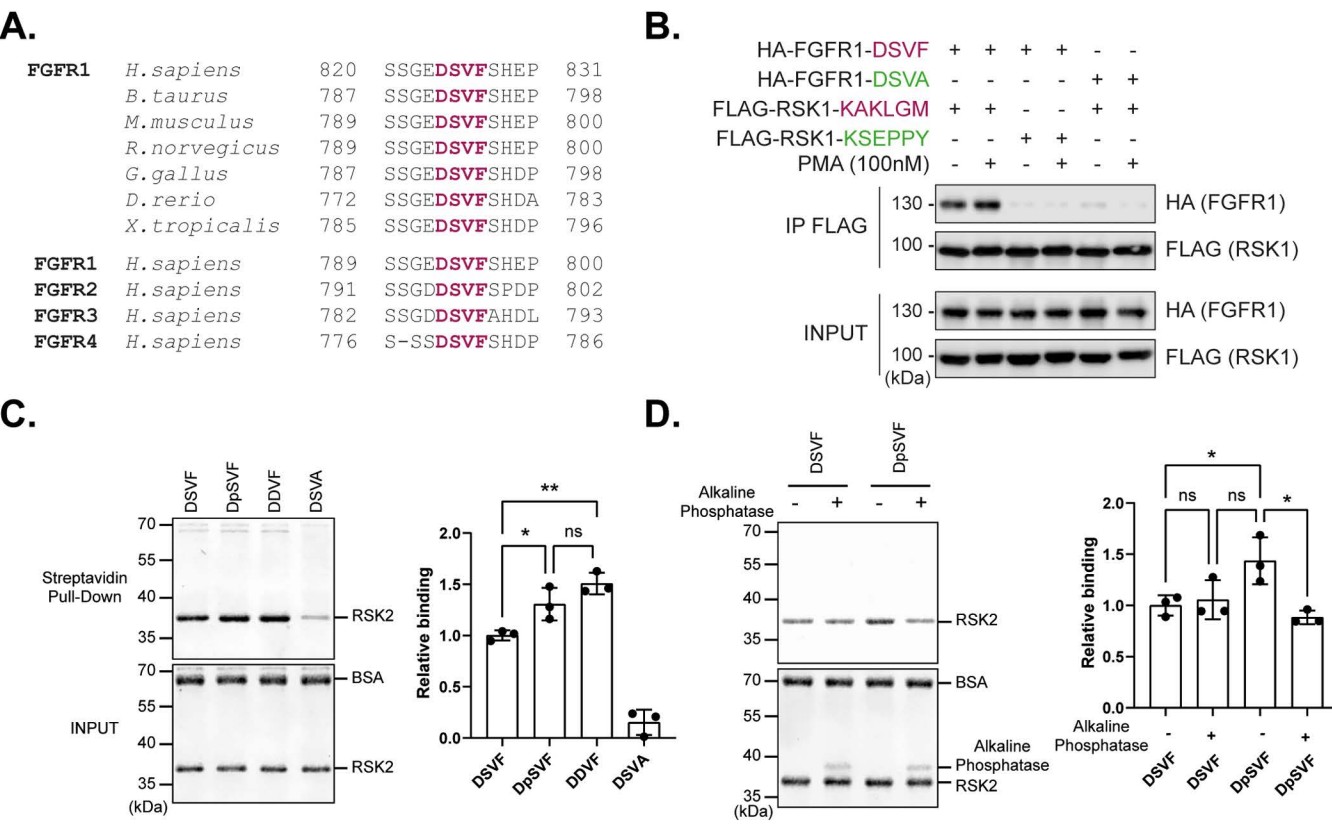

**Fig 2. FGFR1 interacts with RSK1 via a DSVF motif whose phosphorylation increases interaction. A.** The DSVF motif from FGFR1 is highly conserved in all 4 FGFR isoforms and across evolution. **B.** Co-immunoprecipitation of HA-FGFR1 DSVF or the DSVA mutant with FLAG-RSK1 KAKLGM or the KSEPPY mutant, from HeLa cells co-transfected with plasmids expressing indicated proteins. (INPUT = cell lysates; IP = immunoprecipitation), (n = 3). **C.** Assessment of the interaction between biotinylated peptides derived from FGFR1, bearing DSVF, DDVF, DSVA, or DpSVF (phosphoserine) motifs, and purified RSK2 (residues 44-367) using streptavidin pull-down assays (n = 3). **D.** Same experiment as in C comparing the pull-down efficiency of DSVF and DpSVF peptides on RSK2 with or without treatment of the peptides with alkaline phosphatase (n = 3). **C-D.** Quantitative analyses show the mean +/- SD of relative RSK pull-down efficiencies calculated from three independent experiments; P-values: *, p≤ 0.05; **, p≤0.01; ***, p≤0.001.

Interestingly, in the DSVF SLiM found in FGFR1, a serine residue is substituted for glutamate or aspartate negatively charged residues found in pathogens' SLiMs. Since serine phosphorylation would restore a negative charge, we tested if the phosphorylation of the serine in the DSVF motif could increase the DSVF:RSK interaction. In support of this idea, Nadratowska-Wesolowska *et al.* reported a reduced FGFR1:RSK2 interaction after mutation of this serine into an alanine [4].

We thus performed streptavidin pull-down assays using biotinylated synthetic peptides derived from the FGFR1 sequence, with variation in the DSVF motif (DSVF, DpSVF (phosphoserine), DDVF, or DSVA) in the presence of a bacterially expressed, recombinant RSK2 (Fig 2C). As in the case of the full-length FGFR1 protein (Fig 2B), the DSVF-bearing peptide interacted better with RSK2 compared to the DSVA-bearing control peptide (Fig 2C). The DDVF-bearing peptide exhibited even stronger binding to RSK2 than its wild type DSVF counterpart, suggesting that the introduction of a negative charge increased interaction. In line with this, we observed that the phosphopeptide DpSVF displayed a slightly but significantly stronger binding to RSK than the non-phosphorylated DSVF peptide (Fig 2C). Of note, the DpSVF peptide exhibited slightly lower RSK binding ability than the DDVF peptide. This lower binding of the DpSVF peptide to RSK2 might be attributed to the slightly lower purity of the phosphorylated peptide yielded during synthesis when compared to the DDVF peptide, as assessed by routine mass spectrometry (S3 Fig).

To confirm the impact of serine phosphorylation on binding, another set of pull-down assays was performed, comparing pull-down efficacies with and without alkaline phosphatase treatment of the peptides (Fig 2D). Again, a slight but significant difference was observed, showing that de-phosphorylation of the serine residue in the DpSVF phosphopeptide decreased RSK binding. These data suggest the existence of an additional layer of regulation for RSK binding, where kinases could modulate the binding affinity between FGFRs and RSKs through phosphorylation.

## 4. CNKSR2 also interacts with RSK through a DSVF motif

As the three RSK-interacting proteins identified above were known as effectors/regulators of the RAS-ERK MAPK pathway, we wondered whether additional protein from this pathway would use a similar SLiM to interact with RSK. From our 1526 human protein list (Tab B in S1 Table), we selected Connector enhancer of kinase suppressor of ras 2 (CNKSR2) which harbors a well conserved DSVF motif (Fig 3A) predicted to dock in RSK KAKLGM D-site in 5 out of 5 models (S1C Fig and Tab B in S1 Table). A potential RSK binding motif (EDVF) is present at another position in CNKSR1 but absent in CNKSR3 (S2B Fig). Again, co-immunoprecipitation assays performed in HEK293T cells co-transfected with plasmids expressing FLAG-CNKSR2 (DSVF or DSVA mutant) and HA-RSK1 (KAKLGM or KSEPPY mutant) showed that the DSVF motif of CNKSR2 interacts with the KAKLGM site of RSK1 (Fig 3B).

## 5. The RSK SLiM-binding site participates in the regulation of the MAPK pathway and RSK itself

Cellular RSK interactors identified above, such as SPRED2 [28], GAB3 [14], FGFR1 [4], CNKSR2 [29], are related to the RAS-ERK MAP kinase pathway, as does SOS1, which was suggested to interact with the same RSK SLiM-binding site [5,30]. Interaction between these proteins and RSKs might thus be involved in fine tuning the RAS-ERK-MAPK pathway (Fig 4A and 4B). To test this hypothesis, we monitored ERK activation upon basic fibroblast growth factor (bFGF) stimulation in cells expressing either RSK wild-type (WT) or the

KAKLGM-to-KSEPPY mutant. For this investigation, we made use of our previously generated HeLa RSK1/2 double KO cells (RSK-DKO), which minimally express, RSK3 and RSK4 mRNAs, and are thus virtually RSK-KO cells [13]. These cells were transduced with lentiviral vectors expressing either RSK1 WT (KAKLGM) or the KSEPPY mutant (Fig 4C). Upon bFGF stimulation, the phospho-ERK signal remained more elevated in RSK-DKO cells transduced with the empty vector than in cells transduced with the RSK1 WT (KAKLGM) expression vector (Fig 4D and 4E), thus corroborating a role for RSK in the negative feedback of the RAS-MAPK pathway. Furthermore, cells expressing the RSK1 KSEPPY mutant exhibited increased phospho-ERK levels over time compared to cells expressing WT RSK1 (Fig 4D and 4E). This observation suggests that the DDVF-like SLiM-mediated interaction of cellular proteins with RSKs contributes to the overall negative feedback response of the RAS-ERK MAP kinase pathway.

## 6. Pathogen proteins mimicking human DDVF SLiMs compete for RSK binding and modulate the ERK-MAPK pathway

Proteins of pathogens harboring a DDVF-like motif, such as L or YopM, were reported to act as adapter proteins to retarget RSKs toward unconventional substrates such as nucleoporins or pyrin, respectively [32,33]. However, as suggested by Alexa *et al.* [14], pathogens' proteins might also use their DDVF-like motif to compete with human proteins for RSK binding, thereby interfering with RAS-ERK signaling.

Competition between YopM and SPRED2 for RSK binding was assessed using co-immunoprecipitation experiments in HEK293T cells co-transfected with constructs expressing FLAG-SPRED2 and HA-YopM (from *Yersinia enterocolitica*). Wild-type YopM reduced the ability of SPRED2 to co-immunoprecipitate RSK, whereas a DDVA-mutant YopM did not (Fig 5A). Furthermore, wild-type YopM altered SPRED2 phosphorylation. YopM increased RSK phosphorylation, as previously reported [13], and simultaneously decreased SPRED2 phosphorylation (Fig 5B). These observations fit with the model proposed by Alexa *et al.* [14], wherein binding of viral proteins harboring the DDVF motif increases RSK phosphorylation but decreases phosphorylation of DDVF-dependent substrates (such as SPRED2 itself) (Fig 5C).

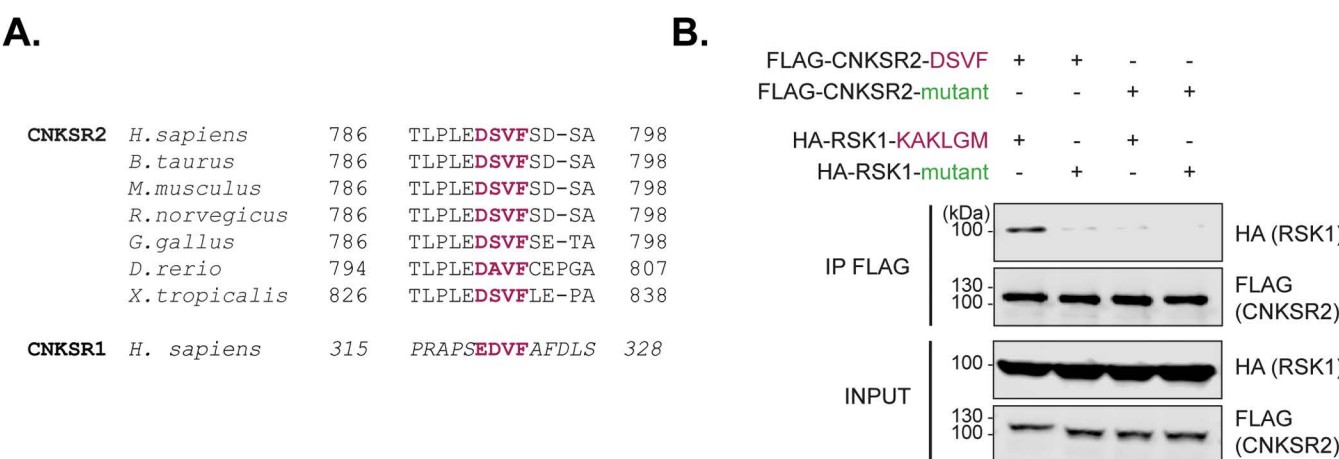

**Fig 3. CNKSR2 interacts with RSK1 through a DSVF motif. A.** The DSVF motif of CNKSR2 is conserved across evolution. Note that a potential RSK binding motif (EDVF) is present at another position in CNKSR1. **B.** Co-immunoprecipitation of FLAG-CNKSR2 DSVF or the DSVA mutant with FLAG-RSK1 KAKLGM or the KSEPPY mutant, from HEK293T cells co-transfected with plasmids expressing indicated proteins (n = 2).

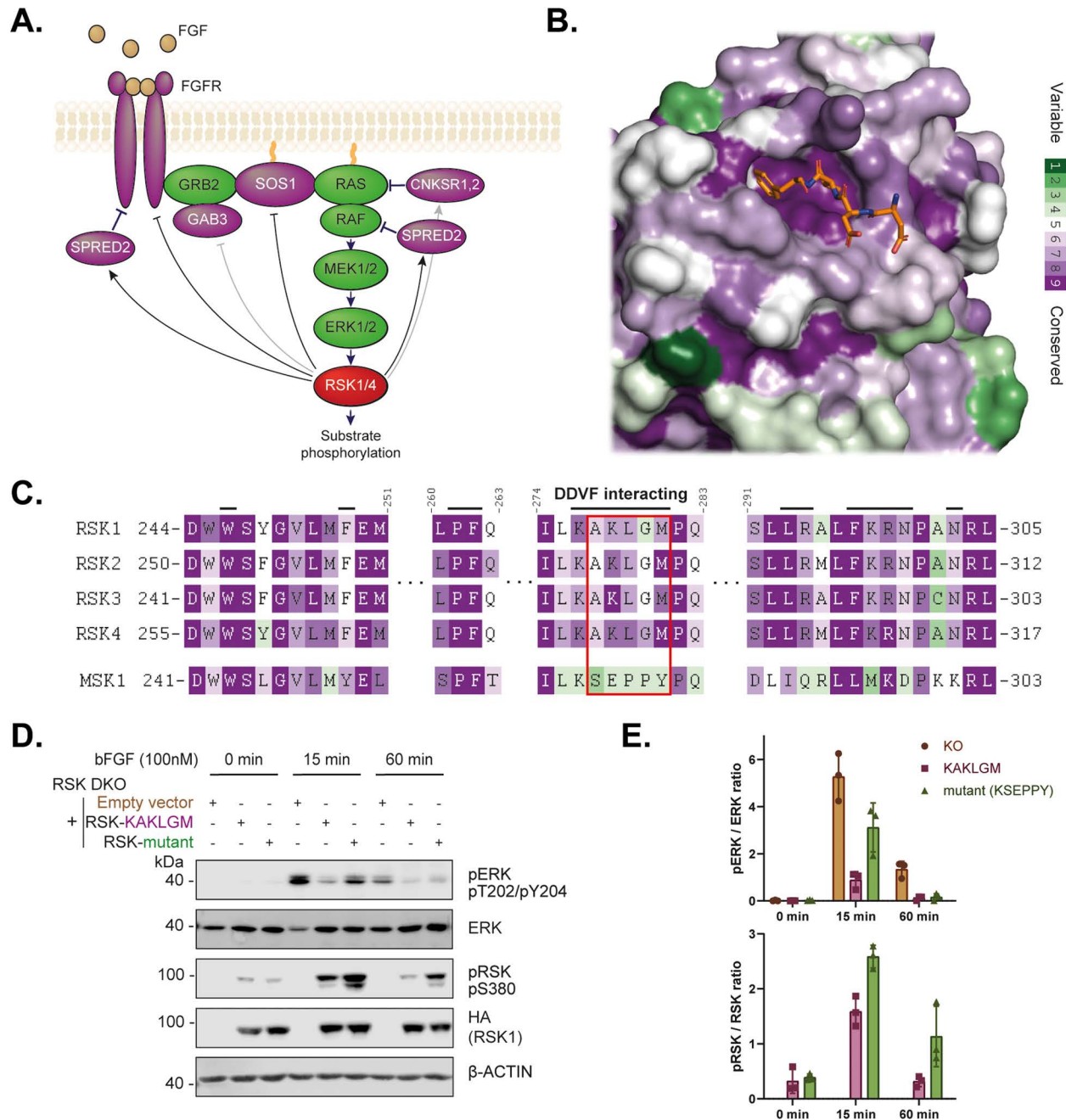

**Fig 4. The conserved DDVF-interacting region contributes to an RSK-mediated negative feedback of the ERK-MAPK pathway in HeLa cells.**
**A.** Several proteins from the RAS-MAPK pathway bear a DDVF-like motif (purple) [31]. In the case of SPRED2, FGFR1 and SOS1, indicated RSK-dependent negative feedbacks were described in [4–6,28]. Black and gray arrows distinguish previously reported RSK interactors from newly identified ones respectively. Black arrows: The SPRED2-RSK interaction was recently demonstrated by Lopez *et al.* [6]. Nadratowska-Wesolowska *et al.* [4] described the FGFR1-RSK2 interaction, though the role of the DDVF motif was not investigated. Póti *et al.* [30] recently reported the SOS1-RSK interaction. To our knowledge, the interactions of RSK with GAB3 and CNKSR2 have not been previously described. **B-C.** RSK tridimensional structure (B) and sequence alignment (C) showing the conservation of the RSK DDVF docking site across evolution (from purple = highly conserved to green = poorly conserved). **B.** RSK2 tridimensional structure colored according to conservation across evolution with the DDVF peptide colored in orange (PDB: 7OPO, [14]). **C.** Sequence alignment showing that the DDVF-interacting residues (indicated by upper black bars) in the KAKLGM region are well conserved across all four RSK isoforms but not in the closely related MSK1, justifying the KAKLGM-to-KSEPPY mutation. **D-E.** Immunoblotting of lysates from RSK-DKO cells transduced with an empty vector (orange), or with vectors expressing RSK1 KAKLGM WT (purple) or the KSEPPY mutant (green). Cells were starved for 14-16 hours and then stimulated with 100nM bFGF for indicated periods of time. Western blots (n = 3) were quantified to calculate phospho-ERK/ERK and phospho-RSK/RSK ratios (mean +/- SD).

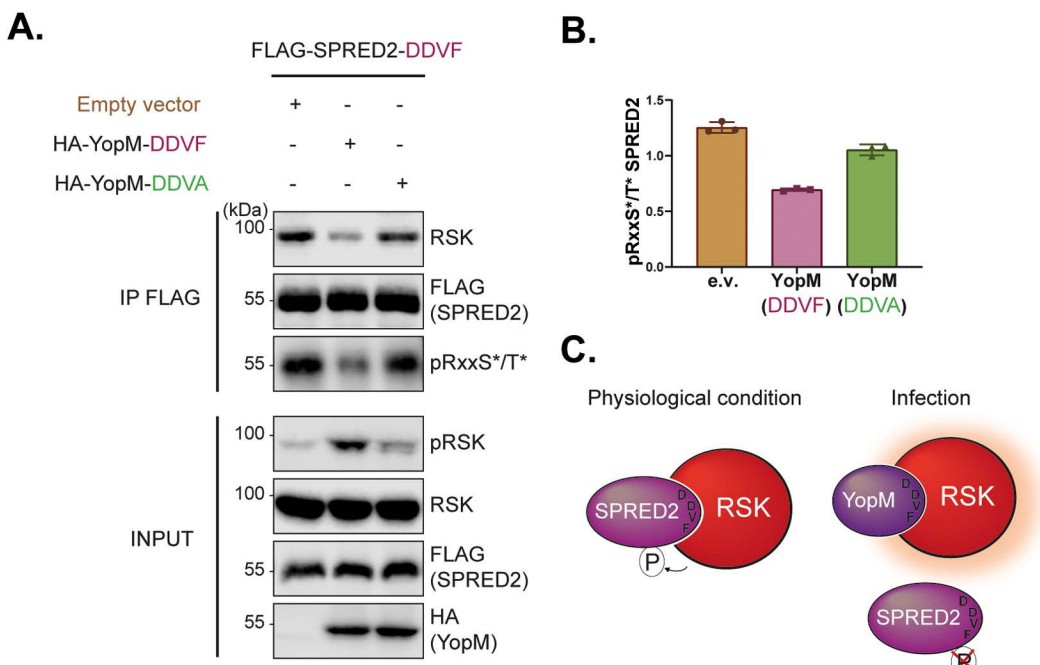

**Fig 5. YopM competes with SPRED2 for RSK binding via the DDVF motif. A.** Immunoblots showing the co-immunoprecipitation of endogenous RSK with FLAG-SPRED2 in the presence or absence of HA-YopM (wild-type or DDVA mutant), which acts as a competitor (n = 3). FLAG-SPRED2 was immunoprecipitated, and co-immunoprecipitation of RSK and phosphorylation of SPRED2 at pRxxS*/T* sites were assessed (*: Phosphorylated residues; x: any amino acid). **B.** Quantification of SPRED2 phosphorylation (pRxxS*/T*) in the presence or absence of HA-YopM, based on data from panel A (n = 3). **C.** Proposed model of YopM-mediated regulation of SPRED2 phosphorylation by RSK. Under physiological conditions, RSK binds to SPRED2 via its DDVF motif, leading to SPRED2 phosphorylation. Upon YopM expression, YopM enhances RSK activation (depicted as increased red shading) while it competes with SPRED2 for RSK binding using its DDVF motif, reducing SPRED2 phosphorylation.

We next tested whether competition occurs in an infection setting. For this purpose, Theiler's murine encephalomyelitis virus (TMEV or Theiler's virus) was used to infect HeLa cells transduced to express either the wild type FLAG-tagged SPRED2 (DDVF) or the SPRED2 DDVA mutant which has decreased affinity for RSK. Cells expressing FLAG-SPRED2 were either mock-infected or infected with TMEV expressing $L^{WT}$ (DDVF) or L mutants with either the DDVA mutation which affects RSK binding, or the M60V mutation which does not prevent RSK binding but inhibits L protein ability to disrupt the nucleo-cytoplasmic trafficking [32]. As shown in Fig 6A, less SPRED2 co-immunoprecipitated with RSK from cells infected with viruses expressing L proteins able to bind RSK ($L^{WT}$ or $L^{M60V}$) than from mock-infected cells or from cells infected with the DDVA mutant.

The observed competition between the L protein and SPRED2 appears relatively weak. This may be attributed to the supraphysiological levels of FLAG-SPRED2 in our assays, which was estimated to be approximately 100-fold higher than physiological SPRED2 levels (S4A and S4B Fig). Such overexpression could partially mask the competitive effects that might occur under more physiologically relevant conditions.

Taken together, our results support the idea that DDVF-containing proteins of pathogens compete with cellular proteins, which bind RSK through a similar SLiM.

We next examined whether competition for RSK binding by pathogen-derived DDVF motifs affected ERK-MAPK signaling. In this experiment, murine L929 cells were used because these cells can be infected by TMEV at high efficiency. Moreover, this infection model

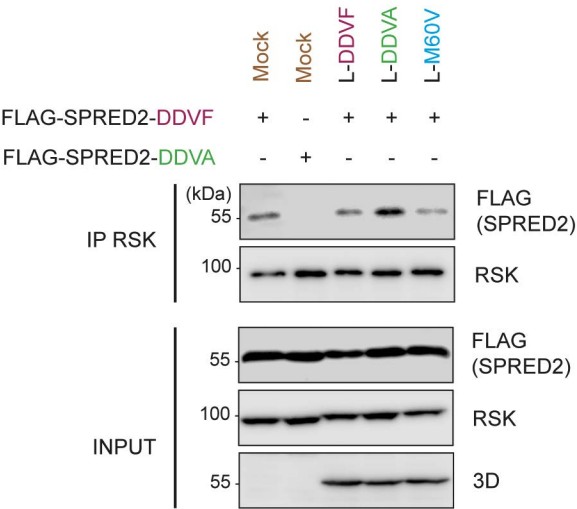

**Fig 6. The L protein competes with SPRED2 for RSK binding via the DDVF motif in TMEV-infected cells.**
Immunoblots showing the co-immunoprecipitation of FLAG-SPRED2 with endogenous RSK in presence or not
of the L competitor (n=2). HeLa cells were transduced with either FLAG-SPRED2(WT) or the FLAG-SPRED2
(DDVA) mutant and infected with TMEV expressing wild-type L (DDVF) or L mutants: DDVA which lost RSK
binding or M60V which conserved RSK binding but lost nucleo-cytoplasmic trafficking perturbation ability.
Viral 3D polymerase served as an infection marker.

is homologous (mouse virus in mouse cells). As shown in Fig 7A and 7B, L protein modulated
ERK phosphorylation in a DDVF-dependent manner as both L variants that bind to RSK ($L^{WT}$
and $L^{M60V}$) increased ERK Thr202/Tyr204 phosphorylation. Our interpretation is that L likely
competed for RSK binding with DDVF-containing proteins like SPRED2, which negatively
regulate the ERK pathway.

We also tested whether RSK activation (pSer380 phosphorylation) correlated with phos-
phorylation of eIF4B, a downstream substrate of RSKs. As shown in Fig 7A, eIF4B phosphory-
lation was strongly increased by viral infection but this phosphorylation was inhibited in cells
infected with the wild type virus although this virus increased phosphorylation of both ERK
and RSK. This suggests that eIF4B phosphorylation was more influenced by nucleocytoplas-
mic alteration than by RSK activation.

Together, these results (Figs 5–7) indicate that pathogen proteins, such as YopM from
*Yersinia* species and the L from TMEV, mimic human proteins harboring the DDVF motif and
therefore compete for RSK binding. By doing so, TMEV modulates the ERK-MAPK pathway,
potentially altering host cell signaling to its advantage (Fig 7C).

## Discussion

SLiMs are emerging as omnipresent motifs which fine tune cellular signaling pathways
through subtle protein-protein interactions. Given their small size (usually 3 to 12 amino
acids) and unstructured nature, SLiMs can easily arise from convergent evolution, making
them appealing targets for pathogenic mimicry, in particular in the case of RNA viruses
[22,23]. Previous studies [13,14,24] have revealed that unrelated pathogens, including RNA
viruses, DNA viruses and bacteria expressed proteins carrying a DDVF-like SLiM enabling
them to interact with the highly conserved KAKLGM SLiM-binding site of RSKs. We pos-
tulated that these pathogens mimic cellular proteins, which bear a similar DDVF motif to
influence RSK activity or subcellular localization through this docking site.

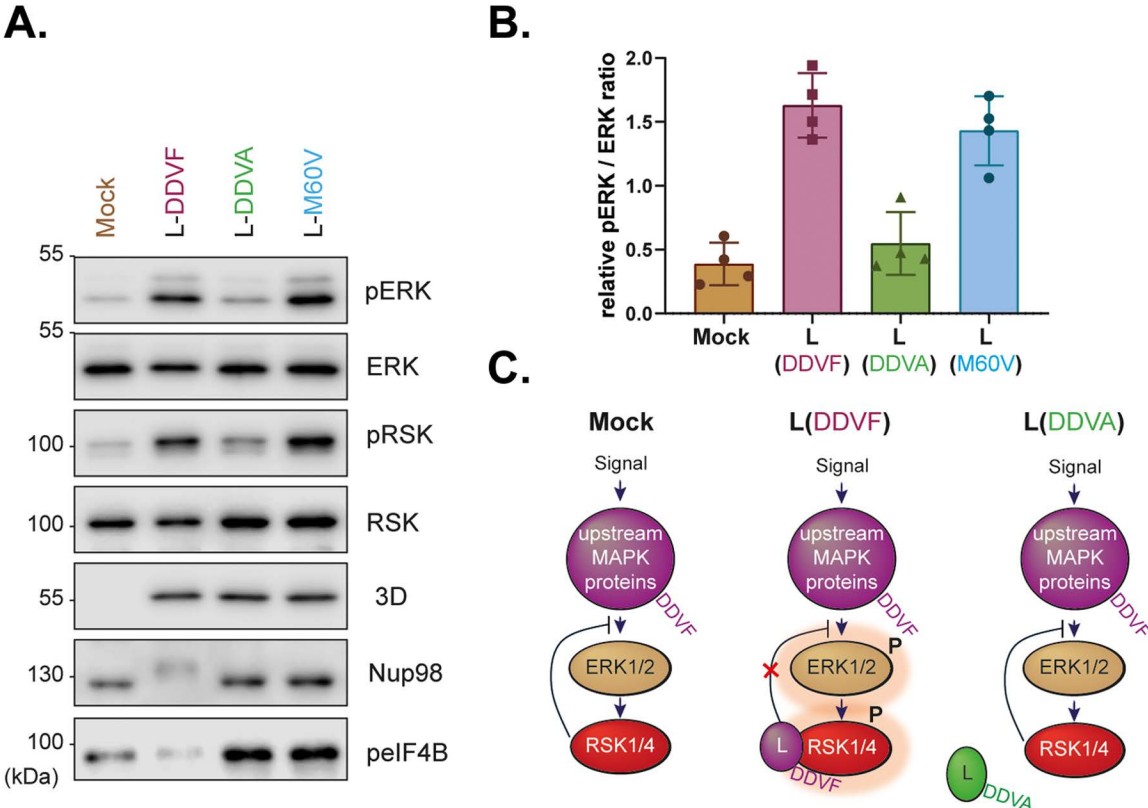

**Fig 7. DDVF-mediated modulation of ERK signaling in TMEV-infected cells. A.** Immunoblot analysis of L929 cells infected with TMEV derivatives expressing either the wild-type L protein (L-DDVF) or its mutants (L-DDVA and L-M60V) compared to uninfected controls (n = 4). Phospho-ERK (pERK) and total ERK levels were assessed. Viral 3D polymerase detection served as an infection control. **B.** Quantification of the pERK/ERK ratio from the immunoblots shown in panel A (n = 4). **C.** Proposed model of RSK-mediated ERK-MAPK regulation by L during TMEV infection. The DDVF motif of L competes with host ERK-MAPK proteins containing DDVF motifs, thereby inhibiting DDVF-mediated negative feedback on the ERK-MAPK pathway. In the absence of a functional DDVF motif in **L** (mutant forms, shown in green), the virus does not dysregulate the MAPK pathway. Proteins in the MAPK pathway with DDVF motifs (e.g., SPRED2, GAB3, CNKSR2, FGFR1, SOS1, etc.) are shown in purple. Activated RSK and ERK kinases are represented by red shading.

To find such regulator proteins, we screened the human proteome for proteins containing an evolutionary conserved DDVF-like motif. AlphaFold-multimer provided an effective support to the candidates' selection pipeline prior to in vivo validation, as the predictions displayed the expected interface in 5 out of 5 models for the 4 confirmed RSK-interacting proteins identified. In all cases, the DDVF-like SLiM of human proteins interacted with the KAKLGM region of RSK in the same way as that proposed for the SLiM of pathogens' proteins. The similarity of the interfaces was structurally confirmed for RSK interaction with Kaposi sarcoma-associated herpes virus protein ORF45 [14] and with the human protein SPRED2 [6].

Our work shows that the DDVF-like SLiM occurring in pathogens' proteins mimics a SLiM likely to be important for fine tuning of the RAS-ERK MAP kinase pathway. This pathway and the downstream RSK kinases phosphorylate a surprisingly large variety of substrates [34] and thereby regulate important biological processes including cell survival, growth and translation. Tight regulation of this pathway is crucial to prevent the disastrous consequences of its over-activation. Previous research has highlighted the role of RSK, SPRED2, as well as RSK targets

such as FGFR1 and SOS1 in negative feedback to the ERK-MAPK pathway [7]. We identified GAB3 and CNKSR2 as new RSK interactors. GAB proteins are recognized for their role in modulating various signaling pathways, including MAP kinase pathways [35]. GAB3, the least studied among the GAB isoforms, is the only one possessing a DDVF-like motif, suggesting that it acts through a mechanism involving RSK (S1B Fig). CNKSR proteins are also known modulators of the RAS-ERK MAP kinase pathway [36]. Both CNKSR isoforms, CNKSR1 and 2, contain motifs (EDVF and DSVF, respectively) that dock with the RSK KAKLGM region, as predicted by AlphaFold.

The RSK interaction motifs detected in FGFR1 and CNKSR2 (DSVF) contain a serine instead of a negatively charged amino acid (Asp or Glu) found in the SLiM of pathogens' proteins. Interestingly, according to its phosphorylation status, the DSVF motif in FGFR1 exhibits a tunable affinity for the electropositive KAKLGM motif of RSK, (Fig 2C and 2D). Although the observed effect of peptide dephosphorylation was rather modest and its physiological importance uncertain, this observation underscores a potential novel layer of RSK regulation, involving other Ser/Thr kinases, which would modulate the temporality or the subcellular localization of RSK association with those proteins, via phosphorylation of the SLiM.

As suggested by our data (Fig 4), the occurrence of DDVF-like SLiMs in these proteins could act to downregulate the RAS-ERK pathway at multiple levels, safeguarding against its overactivation in cells. The importance of the KAKLGM SLiM binding site in the regulation of RSKs' physiological activity likely offered an easy target to pathogens for SLiM mimicry.

By expressing proteins containing a DDVF-like SLiM, pathogens likely compete with endogenous DDVF-bearing proteins for RSK binding, thereby affecting the physiological regulation of the RAS-ERK-RSK pathway. This hypothesis was corroborated by our finding that YopM expression, in a transfection setting (Fig 5) and L expression, in an infection setting (Fig 6) decreased RSK interaction with SPRED2. Moreover, data shown in Fig 7 confirm that infection of cells with viruses that express L proteins which contain the DDVF motif (L^{WT} and L^{M60V}), trigger a more pronounced activation of the ERK pathway. Dysregulation of ERK activity may lead to effects such as increased translation or dysregulated apoptosis, which may benefit the pathogen. Competition by pathogens' proteins is likely an effective mechanism in the case of viral infections given the often-high expression level of viral proteins in the infected cell.

While our findings indicate a DDVF-dependent ERK-MAPK regulation mechanism, the complexity of the ERK-MAPK pathway, which involves multiple regulators and potential indirect effects, prevents us from concluding that the observed effects are solely due to direct competition. The consistent DDVF-dependent modulation of both SPRED2 and ERK phosphorylation however underscores the importance of the DDVF-like motif in pathogen-driven subversion of host signaling pathways.

In addition, as was shown for *Yersinia* and for cardioviruses, interaction of pathogens' proteins with RSKs through SLiM mimicry may be used to retarget RSKs toward unconventional substrates. *Yersinia* YopM was shown to form a complex with several proteins including RSK, another kinase named PRK or PKN, as well as with pyrin, leading to pyrin phosphorylation and ultimately to inflammasome inhibition [33,37]. In the case of cardioviruses, L protein interaction was shown to target RSK toward the nuclear pore complex where RSK phosphorylates FG-nucleoporins such as NUP98, thereby disrupting nucleo-cytoplasmic trafficking in the cell [32]. Pyrin and NUP98 were not detected among the regular RSK substrates [34], suggesting that YopM and L can act as bridging platforms to increase the contact between RSKs and unconventional substrates, thereby promoting their phosphorylation by RSKs.

The interface between DDVF-like SLiMs and RSKs offers a screening opportunity for molecules that modulate RSK functions or prevent their hijacking by pathogens, as shown in a recent screening setup [30] and in a study targeting Kaposi's sarcoma-associated herpesvirus (KSHV) infection [38]. However, it is worth noting that inhibition of the RSK-SLiM interaction in this context may in part compromise the fine regulation of the RAS-MAPK pathway. This might raise concerns in the case of long-term treatments as the RAS-MAPK pathway is often hyperactivated in cancer. In contrast, molecules inhibiting DDVF-RSK interaction might prove useful as short-term treatment to attenuate the severity of acute viral or bacterial infections.

## Materials and methods

### Cells

HEK293T [39], L929 (ECACC ref 85011425), and HeLa M cells, a subclone of HeLa cells kindly provided by R. H. Silverman [40] referred to as HeLa cells, were maintained in Dulbecco's Modified Eagle Medium (Lonza) supplemented with 10% fetal bovine serum (Sigma), 100 U/mL penicillin and 100μg/mL streptomycin (Lonza). RSK1 and RSK2 double knock out HeLa M cells referred to as RSK-DKO cells were generated and validated in [13]. One transcribed RSK1 allele is however detected carrying a 81nt in-frame deletion. These cells express low levels, if any, of RSK3 and RSK4 mRNA and can thus be considered as virtually RSK-KO cells. All cells were cultured at 37°C in a humidified atmosphere containing 5% $CO_2$.

### Plasmids, retroviral and lentiviral constructs

Expression plasmids and retro/lentiviral expression vectors are presented in S2 Table. Note that tagged proteins expressed by these vectors contain either 3xFLAG or 3xHA tags which are referred to as FLAG- and HA- in the text and figures. Plasmid vectors include pcDNA3 (Invitrogen) for mammalian expression. Retroviral expression vectors were derived from pQCXIH (Clontech) and lentiviral vectors were derived from pTM952, a derivative of pCCLsin.PPT.hPGK.GFP.pre [41,42]. Lentiviral constructs expressing human RSKs were constructed using the Gateway technology (Invitrogen) from donor plasmids. pTM1116 encodes Human RSK1 [32]. Donor plasmids encoding Hs.SPRED2 Hs.SPRED1 and HsCNKSR2 were kindly provided by Dominic Esposito through the Addgene collection (Addgene refs: #70573, #70607, #70605 and #70313 respectively), HsGAB3 through the DNASU collection (ref: HsCD00081299) and Hs.FGFR1 was kindly provided by Jean-Baptiste Demoulin. Finally,

pTM1005 encoding for 3xFLAG-YopM (*Y.enterocolitica* W22703) and pTM1020 for 3xFLAG-YopM[F366A] (F366A mutant YopM) were described by Sorgeloos *et al.* [13]. Note that, where indicated, lentiviral or retroviral constructs were transfected as expression plasmids instead of being transduced.

### Viruses

TMEV viruses used in this study are derivatives of KJ06, a variant of the persistent DA strain (DA1 molecular clone) carrying a capsid selected for efficient growth in L929 cells [43]. The FB09 mutant of KJ06 carries the M60V mutation in the Leader protein (L), previously shown to neutralize L-mediated toxicity [44], while the MIP146 mutant [13] contains the F48A mutation that disrupts the interaction between the L protein and RSK. For the experiment shown in Fig 7, the FLAG-tagged Leader protein equivalents of KJ06, FB09, and MIP146 (referred to as TM994, TM1117, and TM1115, respectively) were used (S2 Table).

To generate these viruses, constructs encoding full-length viral genome cDNAs were transcribed in vitro (RiboMax Transcription Kit, Promega). The resulting RNA corresponding to the viral genome was electroporated into BHK-21 cells using a Gene Pulser apparatus (Bio-Rad) set to 1500V, 25 μFd, and no shunt resistance. Supernatants were collected 48–72 hours after electroporation, after the appearance of complete cytopathic effects. Supernatants underwent two to three freeze-thaw cycles, were clarified by centrifugation at 1258 × g for 20 minutes, and then stored at -80°C. Virus titers were determined via plaque assays in BHK-21 cells.

## Infections

To assess the effect of TMEV infection on ERK-MAPK regulation (Fig 7A and 7B), L929 cells were seeded at 40,000 cells per well in 24-well plates (2 plates, 8 wells each) on Day 1. On Day 2, the medium was replaced with serum-free medium 16hours before infection. On Day 3, cells were infected at a multiplicity of 2 PFU per cell, calculated based on 150,000 cells per well. 10mM HEPES was included in the virus mix to prevent pH shock. Each well received 200 μL of the virus mix, and after 1 hour of infection at 37°C, 400 μL of serum-free medium was added. At 8.5 hours post-infection, cells were treated with serum-containing medium for 30 minutes. After treatment, cells were washed once with 1 mL PBS. At 9 hours post-infection, proteins were harvested in 80 μl of Laemmli buffer.

For the competition experiment (Fig 6), HeLa M cells transduced to express SPRED2 (WT or DDVA mutant) were infected with the following modifications. On Day 1, 400,000 cells were seeded per well in 25 cm² plates. On Day 2, the cells were infected for 12 hours at a higher multiplicity of infection (MOI of 5 PFU per cell), based on an estimated 800,000 cells per well. Cells were not serum-starved but, before infection, the wells were washed with 2 mL of serum-free DMEM. The infection mix was added (1 mL per well). 1 hour after infection, 4 mL of DMEM with serum was added per well. On Day 3, cells were lysed and RSK-immunoprecipitation was performed as described below.

## Cell transfection and stimulation

HEK293T or HeLa cells, seeded the day before, were transfected using Lipofectamine 2000 (Thermofisher) according to the manufacturer's instructions with a ratio of 0.5 μg/ 2 μl or 2.5 μg/ 7.5 μl of DNA/transfection reagent for 24-well plates or 6-well plates, respectively. When indicated, cells were stimulated with 100 nM Phorbol-12-myristate-13-acetate (PMA) (Sigma-Aldrich, #P8139) at 37°C for 20 minutes. In the case of bFGF treatments, cells were starved 6-8 hours after transfection for a period of 14-16 hours before stimulation with 100 nM bFGF (Peprotech # 100-18B) and 3.33 nM heparin at indicated timepoints.

## Immunoprecipitations

In the case of RSK immunoprecipitation, 25 μl of Magnetic beads Protein A/G (#88003, Pierce) per condition were washed 3 times in lysis buffer (Tris-HCl 50 mM pH 8, NaCl 150 mM, NP40 0.5%, EDTA 2 mM, PMSF 1 mM and supplemented with protease/phosphatase inhibitors (Pierce)). Washed beads were incubated with 10 μl of RSK antibody (BD610226) for 2 hours at 4°C. Thus formed anti-RSK magnetic beads were washed once with 25μl of lysis buffer per condition and resuspended in 25 μl of lysis buffer before use for immunoprecipitations. For the immunoprecipitations, transfected/infected cells were lysed in lysis buffer and centrifugated at 14,000 x g for 10 min at 4°C. Cleared supernatants were incubated with anti-RSK or anti-FLAG M2 Magnetic Beads (#M8823, Sigma-Aldrich) with gentle agitation for 4 hours at 4°C. Magnetic beads were then washed 3 times with the lysis buffer without inhibitors. Immunoprecipitated proteins were detected by Western-blot analysis.

## Western blot analysis

Total protein extracts were denatured for 5 min in Laemmli buffer at 95 °C, resolved by SDS-PAGE and transferred to polyvinylidene difluoride or nitrocellulose membranes (Immobilon; Millipore). Blocking of unspecific antigens was carried out in 5% nonfat dried milk in Tris-buffered saline (50 mM NaCl, 50 mM Tris-HCl, pH 7.5) for 1 hour at room temperature. Primary antibodies are listed in S3 Table. Reference related to ERK antibody production can be found in [45]. Primary antibodies were diluted in TBST (TBS containing 0.2% Tween 20) and incubated overnight at 4°C with gentle rolling. Membranes were washed three times in TBST for 5 min at room temperature. Species-matched IRDye or HRP secondary antibodies were diluted in blocking buffer as before and incubated at room temperature for 1 h. Membranes were washed again three times in TBST for 5 min at room temperature. Fluorescent signal was detected through an Odyssey Fc infrared imaging system (Li-Cor) while HRP-generated signal was detected using Pierce SuperSignal or Cyanagen Supernova substrates on the same apparatus.

## RSK2 bacterial protein expression and purification

*E. coli* BL21-AI were transformed with pFB42, a plasmid encoding a 6-His N-Terminal fusion with the residues 44-367 of the murine RSK2 under the control of a T7 promoter upstream of the lac operator and constitutively expressing the LacI repressor. A single colony grown on an agar plate containing 250 ug/mL ampicillin was amplified in TSB medium supplemented with the same concentration of ampicillin and further grown under constant shaking at 37°C to an OD of about 0.6. IPTG and arabinose were then added at a final concentration of 1 mM and 0.2%, respectively, and cultures were incubated for another four hours at 37°C under agitation. Bacteria were harvested by centrifugation, resuspended in Tris-HCl 50 mM pH8, NaCl 150 mm, imidazole 20 mM, 10 mg/mL lysozyme, supplemented with protease inhibitors (Roche) and further lysed by sonication. Cleared bacterial lysates were loaded on column packed with 1 mL Ni-NTA agarose (QIAGEN) and washed with 5 column volumes (CV) of Tris-HCl 50 mM pH 8, NaCl 150 mM, imidazole 20 mM, followed by 5 CV of Tris-HCl 50 mM pH 8, NaCl 500 mM, imidazole 20 mM and eluted in 5 CV of Tris-HCl 50 mM pH 8, NaCl 150 mM, imidazole 300 mM. Eluted proteins were dialyzed overnight in Tris-HCl 50 mM pH 8, NaCl 150 mM supplemented with 10% glycerol. Protein purity (>90%) was assessed by SDS-PAGE analysis and Coomassie staining and protein concentration was measured by UV spectrophotometry with the extinction coefficient determined by ExPASy ProtParam [46]. Aliquots of dialyzed RSK2 were snap-frozen and stored at -70°C until further use.

## Biotinylated FGFR1-derived peptide pull-down assay

For RSK2 pull-down assays, 400 pmol of biotinylated FGFR1-derived peptides were incubated in the presence or absence of 5 units of alkaline phosphatase (FastAP, #EF651, ThermoFischer Scientific) at 37°C in 10 mM Tris-HCl, 5 mM $MgCl_2$, 100 mM KCl, 0.02% Triton X-100 and 100 ug/mL BSA in a total volume of 100 uL. Sequences of the biotinylated FGFR1-derived peptides were KDTRSSTCSSGE[DSVF]SHE with the DSVF region replaced by DSVA, DpSVF, DDVF or DSVA amino-acids as indicated. Mass spectrometry analysis confirmed the identity of the synthesized peptides. After one hour, 400 pmol of recombinant RSK2 (44-367) diluted in 100 uL of the same buffer supplemented with 2 mM imidazole and 1 mM $Na_3VO_4$ was mixed with FGFR1-derived biotinylated peptides and incubated for 2 hours at 4 °C under gentle end-to-end agitation. Final concentrations of both RSK2 and the FGFR1-derived peptides were 2 uM. After two hours, 20 uL of a 50% slurry of streptavidin magnetic beads (#88817, Pierce) were added and incubated as before. Bound peptides were then washed three

times with 50 mM Tris-HCl pH 8, 150 mM NaCl, 0.5% NP40, 2 mM EDTA and precipitated RSK2 was eluted in Laemmli buffer and analyzed by SDS-PAGE and Coomassie staining. Gel imaging and protein quantification were performed using an Odyssey Fc imaging system interfaced with Image Studio 5.2 (LI-COR).

## Alignments

Clustal Omega software (https://www.ebi.ac.uk/Tools/msa/clustalo/) was used to produce multiple sequence alignments from human SPRED1-3 (Uniprot sequences Q7Z699, Q7Z698, and Q2MJR0), GAB1-4 (Uniprot sequences Q13480, Q9UQC2, Q8WWW8, and Q2WGN9), FGFR1-4 (Uniprot sequences P11362, P21802, P22607, and P22455), CNKSR1-3 (Uniprot sequences Q969H4, Q8WXI2, and Q6P9H4), and RSK1-4 (Uniprot sequences Q15418, P51812, Q15349, and Q9UK32). To calculate the percentage of conservation of the unstructured C-term of the FGFR1-4 proteins, region annotated as unstructured according to MobiDB [47] (https://mobidb.org/) were aligned with Clustal Omega. Multiple sequence alignments of SPRED2 and FGFR1 from various species were retrieved from the now discontinued Homologene database. Multiple sequence alignments of GAB3 and CNKSR2 from various species was provided by Gene - NCBI (https://www.ncbi.nlm.nih.gov/datasets/gene/). Multiple sequence alignments of RSK1-4 across evolution were given by Proviz [48] (http://proviz.ucd.ie/), and coloring scheme according to conservation was calculated by ConSurf web server (https://consurf.tau.ac.il/).

## AlphaFold-multimer structure predictions and visualization

Protein complex structure predictions were generated on a locally-installed AlphaFold v2.3.2 [19] kindly implemented by Raphael Helaers from the de Duve Institute. Protein complex predictions were performed in the multimer mode using defaults parameters. Inputs were sequences derived from a fragment (amino acid 161-337) of the N-terminal kinase domain from the human RSK2 protein (accession number: P51812) and peptide sequences of 103 amino acid centered on the valine of the D/E-D/E-V-F motif except in situations where the motif was located at the N- or C-terminus of the protein candidates. All structural predictions and associated accuracy metrics have been made available in the zenodo open research data repository (Parts 1 and 2: https://doi.org/10.5281/zenodo.10630296, Parts 3 and 4: https://doi.org/10.5281/zenodo.10653846 and part 5: https://doi.org/10.5281/zenodo.10658284). Multiple structure alignment and visualization were generated using PyMOL 2.5.4 (Schrodinger, LLC).

## Quantification and statistical analysis

Statistical significance was determined using one-way *ANOVA* as implemented in the Prism 8.0.02 statistical analysis software (GraphPad Software, Inc., San Diego, CA). The number of independent experiments (n) and statistical comparison groups are indicated in the figures or figure legends. Asterisks denote the statistical significance of the indicated comparisons as follows: *, $p \leq 0.05$; **, $p \leq 0.01$; ***, $p \leq 0.001$.

## Supporting information

**S1 Table. SLiMSearch and AlphaFold screens.** A. Results of the D/E-D/E-V-F screening using SLiMsearch4, crossed with interaction data available on BioGrid, Intact, PhosphositePlus, and prediction of motif docking within the KAKLGM pocket of RSK using Alphafold-multimer. **B.** Results of the D/E-x-V-F and D/E-V-F screening crossed with

interaction data mentioned hereabove, with AlphaFold-multimer RSK docking predictions for FGFRs and CNKSR2.
(XLSX)

**S2 Table. Plasmids used.**
(XLSX)

**S3 Table. Antibodies used.**
(XLSX)

**S1 Fig. Comparison of SPRED and GAB isoform sequences and AlphaFold docking.** A-B. Conservation of DDVF (purple) motif across all three SPRED (**A**) and GAB (**B**) proteins. **C.** AlphaFold-multimer docks DDVF-like motifs from SPRED2 (green), GAB3 (red), FGFR1-4 (blue), and CNKSR2 (purple) in the RSK KAKLGM docking site. The predicted structures superimpose with that of the crystal structure of ORF45 DDVF (orange) bound to RSK2 (PDB 7OPO).
(TIF)

**S2 Fig. Comparison of FGFR and CNKSR isoform sequences.** A-B. Conservation of the DSVF (purple) motif across all three FGFR (**A**) and CNKSR (**B**) isoforms. Note that CNKSR1 contains an EDVF motif which does not align with the DSVF motif of CNKSR2.
(TIF)

**S3 Fig. Normalized ESI-MS chromatograms showing the mass-to-charge ratios of the FGFR1-derived peptides.** Peptide identities and isotopically averaged theoretical mass-to-charge ratios of doubly protonated peptides are indicated below each chromatogram. Theoretical identities of peptides resulting from incomplete synthesis are marked with an arrow.
(TIF)

**S4 Fig. Comparison of endogenous SPRED2 and transduced FLAG-SPRED2 levels in HelaM cells.** A. Immunoblot showing SPRED2 detection in HeLa M cells transduced with either an empty vector, FLAG-SPRED2 wild-type (WT), or the FLAG-SPRED2 DDVA mutant. **B.** Quantification of the immunoblot in panel A, displaying the calculated fold change between the levels of putative endogenous SPRED2 and the transduced FLAG-SPRED2.
(TIF)

## Acknowledgments

We thank Stéphane Messe for expert technical assistance, Anca Marian for her work as a student, Fabian Borghese for the construction of plasmid pFB42, and Michael Peeters for construction of plasmid pMIP146 carrying the genome of the L$^{F48A}$ mutant TMEV. Finally, we thank Raphaël Helaers for the implementation of the AlphaFold-multimer structure prediction software on the high-performance computing infrastructure of the de Duve Institute.

## Author contributions

**Conceptualization:** Martin Veinstein, Thomas Michiels, Frederic Sorgeloos.

**Formal analysis:** Martin Veinstein, Frederic Sorgeloos.

**Funding acquisition:** Thomas Michiels.

**Investigation:** Martin Veinstein, Vincent Stroobant, Fanny Wavreil, Frederic Sorgeloos.

**Resources:** Vincent Stroobant.

**Supervision:** Thomas Michiels.

**Visualization:** Martin Veinstein, Frederic Sorgeloos.

**Writing – original draft:** Martin Veinstein, Thomas Michiels, Frederic Sorgeloos.

**Writing – review & editing:** Martin Veinstein, Thomas Michiels, Frederic Sorgeloos.

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
