## [Decision Letter · Decision Letter 0]

24 Sep 2024

Dear Prof Michiels,

Thank you very much for submitting your manuscript "The "DDVF" motif used by viral and bacterial proteins to hijack RSK kinases evolved as a mimic of a short linear motif (SLiM) found in proteins related to the RAS-ERK MAP kinase pathway." for consideration at PLOS Pathogens. As with all papers reviewed by the journal, your manuscript was reviewed by members of the editorial board and by several independent reviewers. In light of the reviews (below this email), we would like to invite the resubmission of a significantly-revised version that takes into account the reviewers' comments. An important comment from several reviewers was to include a direct experimental test of your hypothesis.

We cannot make any decision about publication until we have seen the revised manuscript and your response to the reviewers' comments. Your revised manuscript is also likely to be sent to reviewers for further evaluation.

Sincerely,

Nels C. Elde, Ph.D.

Academic Editor

PLOS Pathogens

Ronald Swanstrom

Section Editor

PLOS Pathogens

Michael Malim

Editor-in-Chief

PLOS Pathogens

orcid.org/0000-0002-7699-2064

Reviewer's Responses to Questions

**Part I - Summary**

Reviewer #1: The authors examine Short Linear Motifs (SLiMs) that interact with RSKinases and are used by viruses to potentially dysregulate the RAS-ERK pathway. The authors use co-immunoprecipitation to identify two known and two novel cellular proteins that interact with RSKs using an interface found in proteins encoded by various pathogens. The authors also express a mutated form of ERK that lacks the docking site in HeLa cells, and results suggest that ERK may interact with multiple partners and be involved in a kind of negative feedback. Overall the results are clearly presented and convincing, although ultimately it remains unclear what exactly pathogens gain by modifying RSKs, other than general dysregulation of the RAK-ERK pathway, which is involved in cell cycle regulation and a wide range of other activities.

Reviewer #2: In this manuscript, Veinstein and colleagues identify new proteins that bind p90 ribosomal S6 kinase (RSK) and regulate the RAS-ERK MAP kinase pathway. The authors have previously reported that highly unrelated pathogens, including viruses and bacteria, convergently evolved proteins that hijack host machinery through short linear motifs (SLiMs). The authors previous work suggests that SLiMs with the motif “DDVF” are used by pathogens to bind RSK and alter RAS-ERK MAP signaling. Armed with this knowledge, they hypothesize that similar DDVF motifs likely exist in host proteins and by searching for these motifs they will identify factors that interact with and regulate RSK activity. Here, the authors use SlimSearch4 to identify candidate SLiM-containing proteins and further refined their candidates through an AlphaFold-multimer machine learning structure prediction. The authors identified several members of the MAP kinase pathway that possess DDVF motifs that allow for binding to RSKs. The authors performed biochemical studies to demonstrate interactions between RSK and two known binding partners (FGFR1 and SPRED2), and two unknown partners (GAB3 and CNKSR2) that were predicted from their in silico screen. With this work, the authors have taken a clever approach to identifying new interactions, reinforcing the view that viruses and bacteria are “a window into the cell.” Although their coimmunoprecipitation experiments are well-controlled and suggest that these four proteins interact with RSK through DDVF motifs, the authors have surprisingly not tested whether any viral or bacterial proteins compete with and/or interfere with these interactions to modulate RAS-MAP signaling. Aside from one figure showing changes in phosphorylation of ERK and RSK, it’s also unclear whether these interactions between RSK and DDVF-containing proteins has a meaningful impact on MAPK signaling and cell behavior.

Reviewer #3: In the manuscript “The ‘DDVF’ motif used by viral and bacterial proteins to hijack RSK kinases evolved as a mimic of a short linear motif (SLiM) found in proteins related to the RAS-ERK MAP kinase pathway.”, Veinstein et al. demonstrate a strategy to use pathogen-encoded motifs to identify new, host-host protein interactions using an AlphaFold-based model and then validating with pulldown assays. They also demonstrate that the DDVF motif is interchangeable with a DpSVF motif that the authors hypothesize may be used to regulate interactions with (some?) host binding partners. While similar to other alphafold guided techniques, the use of a pathogen protein as “bait” is an interesting approach, and potentially identifies an important area of host-pathogen interaction regulating the cell cycle. However, this study seems premature, as they do not demonstrate whether the pathogen encoded SLiMs they previously identified can compete with the native motifs to regulate the ERK-MAPK pathway, or that these phosphorylation events do regulate downstream events in the host relevant to host-pathogen interactions.

**Part II – Major Issues: Key Experiments Required for Acceptance**

Reviewer #1: Are HeLa cells the best choice for studying RSK-mediated negative feedback? I understand that they are useful for many applications, but they are also heavily mutated and behave irregularly, in particular with regards to cell cycle control. Would it be worth providing some discussion of this in the text to put it into context? The results seem logical, but it is unclear to me how much they can be extended to real-world infection dynamics.

Reviewer #2: The most conspicuous gap in this study is that the authors have not experimentally addressed the layer of regulation by pathogens. If these pathways and interactions are exploited by viruses and bacteria, it would be important to demonstrate that introduction of at least one protein from a pathogen interferes with the feedback loop on RSK – after all, the PLOS Pathogens audience is interested in pathogens. Citation of previous work demonstrating how viral and bacterial proteins hijack the RSK family through SLiM mimicry is insufficient because readers will need to appreciate the extent to which this pathway and the interactions between the proteins described in this manuscript can be manipulated during infection (for example, by Kaposi sarcoma-associated herpes virus protein ORF45).

The authors should:

1. Perform competition assays to see if a pathogen-associated protein can block the interaction between RSK and DDVF-containing proteins (such as with SPRED2 and GAB3 in Fig 1).

2. Perform a signaling experiment as in Fig 4D-E that includes a pathogen-associated protein as a comparison. For example, wild type cells should be compared head-to-head with cells expressing ORF45, and directly compared to the RSK knockout, RSK-KAKLGM, and RSK-KSEPPY.

3. It would also be helpful if the authors could demonstrate that interrupting these DDVF-RSK interactions has a measurable effect on some aspect of cell behavior (downstream of pERK). Does this actually impact cell survival, growth, and translation to the benefit of a virus or bacteria?

Reviewer #3: 1. The authors make the claim in several locations (eg lines 40-42 and in discussion) “[Previously described pathogen motif bearing proteins] …also likely compete with human proteins”. However, they do not provide any experiments to test this hypothesis. It is important to determine whether the pathogen-encoded DDVF SLiMs do in fact inhibit, modulate, or otherwise interfere with host DDVF-KAKLGM interactions, especially for a pathogen-focused journal. Similarly, some of the previously described pathogen-encoded proteins should be included as controls, eg in Fig. 1 one or more pathogen SLiM-encoding protein should be included as a control to determine whether there may be any difference in binding affinity, etc between host and pathogen proteins.

2. Fig. 2C (Line 179): If purity is an issue with these peptides, then this needs to be discussed. As stated in their manuscript, it’s unclear what the contaminants are, and as such is difficult to assess these blots and their accuracy. Is this non-specific binding, or is this intended to reflect incomplete phosphorylation of the synthetic peptide? In which case how much of the peptide was phosphorylated? Similarly, Fig 2D. “Routine mass spectrometry” is mentioned and the data should be included.

3. Fig. 4: This is unconvincing for the role of the DDVF SLiM. The cellular model tests KAKLGM, which while the authors have shown this interacts with (some) DDVF SLiM proteins, they do not show the inverse, that only DDVF proteins interact with this KAKLGM, or that it is the DDVF-KAKLGM interaction mediating the phenotype they report. The authors also state that the SLiM binding site regulates RSKs’ physiological activity, while only the phosphorylation of RSK and ERK were measured, lacking information on downstream RSK substrates.

**Part III – Minor Issues: Editorial and Data Presentation Modifications**

Reviewer #1: Line 420 – Clustal Omega likely works fine for these applications, and in many cases it is possible to manually curate the short alignments anyway, but in the future the authors may wish to consider using the newest Muscle5 algorithm, which substantially outperforms clustal omega.

For AlphaFold multimer results, what are the pLDDT values? I am curious to know if the authors had a cutoff that they used for these, or how they assessed the quality of the structural predictions. I understand that some of these results are available in the Zenodo repository provided, but given the importance of these values for assessing the results I believe they should be reported in the text.

Reviewer #2: 1. Lines 106-108 are not easy to understand. Could the authors please rephrase to be clearer?

2. PMA treatment in Figure 2 was not explained in the text. Why was PMA used? What were the authors expecting? Is the absence of any obvious change in the pulldowns +/ PMA surprising?

3. Phosphorylation of FGF1R peptide increasing binding by only 25% (and it’s not particularly obvious in Fig 2C). Is this difference in binding physiologically relevant? Could the authors please discuss this and comment on whether this seemingly small effect is meaningful in the context of RAS-MAPK activation?

4. With the model figure in Fig 4A, it would be helpful if the authors could somehow denote previously known interactions and/or label the new biological relationships described in this manuscript. It’s unclear from the figure whether any of these interactions and feedback mechanisms are supported by data from this manuscript (other than the purple shading of proteins with DDVF motifs).

5. It remains difficult to discern how much relative influence the proteins with DDVF motifs have on the negative regulation of RSK. For example, the KAKLGM-to-KSEPPY mutation reduces the inhibitory capacity of RSK on the pERK/ERK ratio by roughly 2- to 3-fold at the 15 min time point (Fig 4E, top panel). How does this level of pERK/ERK ratio compare to the same experiment (stimulation with bFGF) if done in a SPRED2 knockout? A GAB3 knockout? A CNKSR2 knockout? If you observe more activation in knockout cells than with the KAKLGM-to-KSEPPY RSK mutation, that would tell us that there are likely other domains and/or unknown interactions with these proteins that are more important regulating the RAS-MAPK pathway than the DDVF-KAKLGM interaction. Alternatively, if you see the same level of activation in the knockouts (and no further increase in pERK/ERK with the KAKLGM-to-KSEPPY mutant) then it would suggest that most or all regulation is occurring through the DDVF-KAKLGM interaction, which would be an important result.

Reviewer #3: 1. Fig. 1C and 1D: It is inaccurate to claim that RSK1 didn’t IP with SPRED1 as a band can be seen on the figure. SPRED2 is substantially higher, but SPRED1 still IPs better than the SPRED2 mutant or RSK1 mutant. The GAB3 mutant also IPs with RISK1.

2. The authors state in multiple places that pathogens evolved to mimic this human motif, but evolutionary analysis is limited to alignments suggesting the motif is well conserved. There’s no evolutionary analysis to support the claim that pathogens acquired this motif via mimicry evolution. The authors should soften their conclusions, as this is a difficult conclusion to demonstrate. Similarly, a discussion of why different species were chosen in these alignments for Figs 1A, 1B, 2A and 3A. If some of these genes aren’t encoded by the same set of species, that is relevant information about how conserved these motifs may be.

3. Full-length alignments of FGFR isoforms and CNKSR isoforms should be included in the supplemental figures as it was done with SPRED and GAB.

4. Fig. 2B: The purpose and the result of the PMA induction were never discussed in the paper.

5. Line 232: Need to define bFGF.

6. Table S1: summary - the description of column “AFm_Dock” is the same as “DDVF dock”. Table S1B doesn’t have AFm_Dock data. The information in S1C and S1D appears to have already been included in S1A and S1B, respectively.

7. Line 287: Typo. CNKSR2 contains a DSVF motif.

PLOS authors have the option to publish the peer review history of their article (what does this mean? ). If published, this will include your full peer review and any attached files.

**Do you want your identity to be public for this peer review?** For information about this choice, including consent withdrawal, please see our Privacy Policy .

Reviewer #1: No

Reviewer #2: No

Reviewer #3: No
---

## [Decision Letter · Decision Letter 1]

3 Mar 2025

Dear Prof Michiels,

We are pleased to inform you that your manuscript 'The "DDVF" motif used by viral and bacterial proteins to hijack RSK kinases mimics a short linear motif (SLiM) found in proteins related to the RAS-ERK MAP kinase pathway.' has been provisionally accepted for publication in PLOS Pathogens.

Best regards,

Nels C. Elde, Ph.D.

Academic Editor

PLOS Pathogens

Ronald Swanstrom

Section Editor

PLOS Pathogens

Sumita Bhaduri-McIntosh

Editor-in-Chief

PLOS Pathogens

orcid.org/0000-0003-2946-9497

Michael Malim

Editor-in-Chief

PLOS Pathogens

orcid.org/0000-0002-7699-2064

Reviewer Comments (if any, and for reference):

Reviewer's Responses to Questions

**Part I - Summary**

Reviewer #1: The authors have addressed my concerns, performed additional experiments, and added appropriate caveats to the text.

Reviewer #2: The authors have addressed all of my concerns.

Reviewer #3: In this resubmission, Veinstein et al have made substantial changes including 3 new figures that demonstrate the material effect of pathogen encoded DDVF motifs in regulation of the ERK pathway. The competition of YopM is convincing, and while the inhibition of SPRED2-RSK interaction by TMEV L is less impressive, the authors address potential reasons for the difference, and lay the groundwork for future studies testing their hypothesis. With this new work the authors have demonstrated pathogen relevance and responded adequately to reviewer suggestions after their initial submission.

**Part II – Major Issues: Key Experiments Required for Acceptance**

Reviewer #1: (No Response)

Reviewer #2: (No Response)

Reviewer #3: None noted.

**Part III – Minor Issues: Editorial and Data Presentation Modifications**

Reviewer #1: (No Response)

Reviewer #2: (No Response)

Reviewer #3: Lines 135-6 “WT but almost not mutantRSK1…” is worded awkwardly

Line 278 - May want to specify “TMEV” L

PLOS authors have the option to publish the peer review history of their article (what does this mean? ). If published, this will include your full peer review and any attached files.

**Do you want your identity to be public for this peer review?** For information about this choice, including consent withdrawal, please see our Privacy Policy .

Reviewer #1: No

Reviewer #2: No

Reviewer #3: No

---

## [Editor Report · Acceptance letter]

Dear Prof Michiels,

We are delighted to inform you that your manuscript, "The "DDVF" motif used by viral and bacterial proteins to hijack RSK kinases mimics a short linear motif (SLiM) found in proteins related to the RAS-ERK MAP kinase pathway.," has been formally accepted for publication in PLOS Pathogens.

Best regards,

Sumita Bhaduri-McIntosh

Editor-in-Chief

PLOS Pathogens

orcid.org/0000-0003-2946-9497

Michael Malim

Editor-in-Chief

PLOS Pathogens

orcid.org/0000-0002-7699-2064